# Can Transcranial Direct Current Stimulation Enhance Functionality in Older Adults? A Systematic Review

**DOI:** 10.3390/jcm10132981

**Published:** 2021-07-03

**Authors:** Andrés Pino-Esteban, Álvaro Megía-García, David Martín-Caro Álvarez, Hector Beltran-Alacreu, Juan Avendaño-Coy, Julio Gómez-Soriano, Diego Serrano-Muñoz

**Affiliations:** 1Toledo Physiotherapy Research Group (GIFTO), Faculty of Physiotherapy and Nursing, Castilla La Mancha University, 13001 Toledo, Spain; andres.pino36@gmail.com (A.P.-E.); david.martincaro@uclm.es (D.M.-C.Á.); hector.beltran@uclm.es (H.B.-A.); juan.avendano@uclm.es (J.A.-C.); Julio.Soriano@uclm.es (J.G.-S.); diego.serrano@uclm.es (D.S.-M.); 2Biomechanical and Technical Aids Unit, National Hospital for Paraplegia, SESCAM, 45071 Toledo, Spain

**Keywords:** transcranial direct current stimulation, brain stimulation, ageing, motor function, cognitive function

## Abstract

Transcranial direct current stimulation (tDCS) is a non-invasive, easy to administer, well-tolerated, and safe technique capable of affecting brain excitability, both at the cortical and cerebellum levels. However, its effectiveness has not been sufficiently assessed in all population segments or clinical applications. This systematic review aimed at compiling and summarizing the currently available scientific evidence about the effect of tDCS on functionality in older adults over 60 years of age. A search of databases was conducted to find randomized clinical trials that applied tDCS versus sham stimulation in the above-mentioned population. No limits were established in terms of date of publication. A total of 237 trials were found, of which 24 met the inclusion criteria. Finally, nine studies were analyzed, including 260 healthy subjects with average age between 61.0 and 85.8 years. Seven of the nine included studies reported superior improvements in functionality variables following the application of tDCS compared to sham stimulation. Anodal tDCS applied over the motor cortex may be an effective technique for improving balance and posture control in healthy older adults. However, further high-quality randomized controlled trials are required to determine the most effective protocols and to clarify potential benefits for older adults.

## 1. Introduction

Transcranial direct current stimulation (tDCS) is a non-invasive tool for neuromodulation that has proven to be well-tolerated and safe [1,2]. This technique employs low-intensity (1–2 mA) continuous or galvanic current applied transcutaneously via electrodes placed on the scalp [3]. The change generated in the electric potential of the membrane of the underlying neurons affects neuronal excitability, which varies depending on the orientation of the electric field determined by the position and polarity of the electrodes [3,4]. This effect on excitability is believed to be related to transient changes in the synaptic efficiency of different neurotransmitters [4]. In terms of polarity effects by the current, the anode provokes membrane depolarization, which may increase the excitability of cortex neurons. On the other hand, the cathode is capable of hyperpolarizing the neuronal membrane, generating an opposite effect (inhibition) to anode stimulation. Therefore, tDCS may facilitate motor performance by regulating neuron activity in the underlying brain tissue [3]. Several studies have observed that the sole application of a 20-min session of tDCS applied on the dorsolateral prefrontal cortex improve cognitive and motor functions including working memory [5], problem solving [6], decision making [7,8], and movement precision during reaching tasks [9].

Due to demographic changes and improvements in the quality of health services [10], the percentage of population of advanced age has increased considerably in recent years. Complex structural and functional changes in the brain are some of the processes related to normal aging that entail deterioration of cognitive, perception, and motor capacities, which affects daily life activities, independence, and quality of life [11]. The main finding observed is the increase in dual-task costs, and the most affected ability due to aging is the simultaneous execution of one motor and one cognitive task [12,13]. Additionally, older adults present a reduction in the structural and functional plasticity of the brain [14] and in flexibility for tasks requiring previous learning [15]. Trials using neuroimaging indicate that the left dorsolateral prefrontal cortex (DLPFC), which intervenes in the executing function, is one of the key brain regions involved in performing combined cognitive and motor tasks under dual-task conditions [16]. For this reason, tDCS interventions designed for facilitating the functional activation of the DLPFC and its neuronal networks could improve the cognitive function and motor performance in the elderly.

On the other hand, the scientific literature indicates that 50% of elders in nursing homes suffer at least one fall per year [17,18], which can lead to losing independence, serious injuries, and even death [19]. Some studies [20,21,22] have reported that long periods of balance training can improve posture stability and balance in older adults. However, difficulties in achieving sufficient adherence and the presence of fatigue hinder the completion of the extensive training programs required for improving functionality [20,21,22]. Therefore, combining tDCS with exercising programs could enhance motor performance, impacting functionality in the elderly [3]. In general, tDCS can be a viable tool for tackling motor and cognitive deficits related to aging, as it is a safe non-invasive neuromodulation technique.

The main objective of this review was to compile and summarize the currently existing evidence about the use of tDCS for improving diverse functionality variables in older adults, such as strength, balance, posture control, fine manual dexterity, and dual task execution. Additionally, this work intended to determine the optimal stimulation protocols for addressing different alterations of functionality related to aging.

## 2. Materials and Methods

This systematic review was performed according to the guidelines included in the Preferred Reporting Items for Systematic Reviews and Meta-Analyses (PRISMA).

### 2.1. Study Inclusion Criteria

The inclusion criteria were: (1) randomized controlled clinical trials published in English or Spanish, (2) a study population of subjects over 60 years of age with no relevant pathologies, (3) including at least one group receiving brain tDCS, (4) assessed variables including functionality outcomes. The following exclusion criteria was applied: (1) a study population including hospitalized patients due to any medical condition, cardiovascular disease, musculoskeletal disorders, or neurological disorders, (2) studies not specifying tDCS parameters (intensity, placement of electrodes, and session duration).

### 2.2. Literature Databases and Search Strategy

The search for scientific articles was carried out through the following databases: PubMed, Physiotherapy Evidence Database (PEDro), CINAHL, Google Scholar, ENFISPO, and Cochrane Library. The strategy included the following terms: “Transcranial Direct Current Stimulation”, “tDCS”, “older”, “elder”, and “function”, as well as their various synonyms and combinations. The MeSH terms “Transcranial Direct Current Stimulation” were also entered in the PubMed and Cochrane databases. The filters used were randomized controlled clinical trial, human, English and Spanish.

Additionally, an inverse manual search of the references in the articles found was performed. Two independent researchers (APE and DMC) conducted the search separately, and a third researcher (DSM) resolved the discrepancies between both reviewers. No limits were established in terms of date of publication, so the searches included all articles since the creation of the databases up to the 31 August 2020.

### 2.3. Study Selection and Data Extraction

Studies were included for analysis after reading the content of the titles and abstracts. Duplicate articles and studies that did not meet the inclusion criteria were excluded. The full text of the selected studies was then read and those which met the inclusion criteria were evaluated for their eligibility and finally chosen for the qualitative analysis. Discrepancies regarding the inclusion of the final studies by the reviewers were resolved through a discussion/consensus process moderated by a third reviewer. The data described in the results section of each study were extracted using a Cochrane protocol guideline for systematic reviews that ensured the analysis of the most relevant information [23]. Specifically, data measures that are used to verify the results and effects of a treatment should be expressed as mean difference, effect sizes, and interquartile ranges. In addition, the reviewers of this study specified that at least one or more variables should be included related to dual-tasking, cognitive function/task, musculoskeletal performance..

### 2.4. Quality Assessment

Methodological quality of the studies selected in this systematic review was reviewed and assigned according to a series of elements described in the Physiotherapy Evidence Database (PEDro), the reliability of which has been previously reported [24].

The 11 PEDRO elements include: (1) Specified eligibility criteria; (2) Random assign-ment/random distribution of subjects; (3) Concealed allocation; (4) Basal intergroup sim-ilarities; (5) Blinding of subjects; (6) Blinding of therapists; (7) Blinding of evaluators; (8) Follow-up measures of at least one key outcome obtained in more than 85% of initially assigned subjects; (9) Intention-to-treat analysis; (10) Between-group statistical compar-ison; (11) Point and variability measures. 

The PEDRO ranking score for each element was: Yes (1 point); No (0 points); and Do not know (0 points). The total PEDro scale score ranged from 0 to 10 points. The result of each of the selected articles provided an indicator of quality, which could be then be described as excellent (9–10 points), good (6–8 points), fair (4–5 points) or poor (<4 points).

## 3. Results

A total of 237 articles were found in the different databases. After eliminating duplicates and reading the titles and abstracts, 31 articles were selected to be fully read. Finally, nine articles [25,26,27,28,29,30,31,32,33] met the criteria to be included in this systematic review. Figure 1 displays the outcomes of each stage of the selection process. Table 1 shows the results of the methodological quality assessment. The most relevant characteristics of the included articles are described in Table 2.

### 3.1. Characteristics of Participants

The sample of this review was comprised of 260 subjects (49.8% men and 50.2% women), with ages between 61 and 85 years. Of them, 255 participants completed their relevant intervention protocols with tDCS. 

### 3.2. Stimulation Patterns and Parameters 

Placement of electrodes varied among trials, even if all employed anodal tDCS (a-tDCS) stimulation. Three trials [28,29,32] applied a-tDCS over the left prefrontal cortex [34] with the cathode placed on the right supraorbital region. Ljubisavljevic et al. [27] assessed the effect of different electrode placement combinations on the DLPFC: one group received a-tDCS unilaterally on the DLPFC with the cathode on the contralateral supraorbital region, while the other group received tDCS bilaterally over the DLPFC. On the other hand, Oki et al. [30] placed the anode on the right primary motor cortex (M1) at the region corresponding to the biceps brachii (as identified by transcranial magnetic stimulation) while the cathode was placed on the left supraorbital region. Kaminski et al. [26] located the anode on the M1 bilaterally, specifically at the region of the leg, and the cathode on the right supraorbital area. The studies by Baharlouei et al. [25] and Yosephi et al. [31] placed the anode on the left M1 and the cathode on the right supraorbital region. Zhou et al. 2018 [33] set the anode on the left sensorimotor cortex and the cathode on the right supraorbital margin. See Table 2.

In terms of electrode size and current density, most trials employed 35 cm^2^ electrodes [27,28,29,30,31,32,33] and 0.06 mA/cm^2^, except for Oki et al. [30] and Ljubisavljevic et al. [27], who applied a density of 0.04 mA/cm^2^ for the same electrode size. Moreover, Baharlouei et al. [25] used 27 cm^2^ electrodes for the anode (0.07 mA/cm^2^) and 36 cm^2^ for the cathode (0.05 mA/cm^2^), and Kaminski et al. [26] employed electrode sizes of 25 cm^2^ for the anode (0.04 mA/cm^2^) and 50 cm^2^ for the cathode (0.02 mA/cm^2^). On the other hand, the duration of sessions for applying tDCS showed very low variability, with all trials applying the treatment for 20 min each time, except for 30-min sessions in Ljubisavljevic et al. [27].

The protocol selected to apply sham stimulation differed among studies, although all conducted a ramp-up until reaching the target intensity and a ramp-down to 0.0 mA, which was maintained for the rest of the session. Baharlouei et al. [25] and Kaminski et al. [26] applied a 30-s ramp-up to reach the aimed intensity and a 30-s ramp-down immediately afterward. Manor et al. 2016 [29] employed a 30-s ramp-up followed by a 60-s ramp-down. Zhou et al., 2015 [32] increased the intensity manually by 0.1 mA intervals until reaching 2mA, maintained the stimulation for 60 s, and lowered it to 0.0 mA. Ljubisavljevic et al. [27] employed a 30-s ramp-up at the beginning and a 30-s ramp-down at the end of the session. Manor et al. 2018 [28] used a 60-s ramp-up and ramp-down. Yosephi et al. [31] increased the intensity automatically for 10 s up to reaching 2 mA, maintained the intensity during 30 s, and lowered it over 10 s. Zhou et al. 2018 [33] set a 60-s ramp-up to reach the desired intensity followed by 60-s of maintained current and a 60-s ramp-down. Finally, Oki et al. [30] included a ramp-up and ramp-down process during the first 30 s of stimulation. 

### 3.3. Number of Sessions and Duration of Treatment

Five studies [25,27,29,32,33] conducted a crossover trial with two stimulation sessions and a washout period of 1–2 weeks. Two studies carried out a protocol of repeated tDCS sessions: Yosephi et al. [31] applied a two-week treatment with three weekly sessions, and Manor et al. 2018 [28] completed ten sessions over two weeks. On the other hand, the trial by Kaminski et al. [26] comprised two consecutive training sessions, with a 24-h washout in between. As an exception, Oki et al., 2019 [30] designed a protocol of three training sessions where tDCS was only applied in the third one.

### 3.4. Recorded Variables and tDCS Effect

Table 3 showed the main results of the studies included in this systematic review. Most trials assessed subjects immediately after each session, independently of whether two or more tDCS intervention sessions were conducted [25,26,29,30,31,32,33]. Additionally, Ljubisavljevic et al. [27] evaluated the participants during the tDCS stimulation. Finally, the study by Manor et al. 2018 [28] was the only one to include follow-ups, conducted at 2 days and 2 weeks after the intervention. 

In terms of assessed variables, seven studies [25,26,28,29,31,32,33] included the evaluation of balance. Five trials [25,26,28,31,32] evaluated stability while standing using posturography, two studies measured stability while walking using the “Timed Up-and-Go” (TUG) test [28,33] and another one [31] employed the Berg Balance Score (BBS) scale to assess balance while performing different daily life activities. On the other hand, three trials [25,28,29] recorded space–time parameters of gait such as speed, length of stride, and sway of the torso. Additionally, these variables were recorded under dual-tasking conditions in four studies with the aim of assessing the efficiency of execution after adding a cognitive task [27,28,29,32]. Dual task costs were calculated for each outcome as follows: [(performance of dual task—performance of single task)/performance of single task] × 100. Two studies evaluated the cognitive function related to dual tasking via the “Serial Sevens Subtraction Test” (SSST) [27,29,32]. The performance of the cognitive task was measured through the percentage of correct guesses in the SSST and the Montreal Cognitive Assessment (MoCA) [28].

A single study evaluated fine manual dexterity during dual tasking (manual dexterity and cognitive task) [27], for which the Grooved Pegboard Test (GPT) was used. This study also analyzed the general effects of tDCS on motor performance via the simple reaction time (SRT). Finally, one trial [30] evaluated the strength of biceps brachii using dynamometry and muscle activity during maximum voluntary contraction as measured by surface electromyography, with both measurements taken after a single tDCS session. 

#### 3.4.1. Effect on Static and Dynamic Stability

All the included studies [25,26,28,29,31,32,33] that assessed static and dynamic stability found significant differences for the improvement of any of the recorded variables following the tDCS intervention when compared to sham stimulation. However, the study by Kaminski et al. [26] did not find significant differences between groups in the time in balance (TiB) outcome. 

Baharlouei et al. [25] observed a significant interaction between time and stimulation, showing that active stimulation (a-tDCS applied over M1) could improve balance indices, whereas sham stimulation had no effect on balance in older adults. This was evidenced in the a-tDCS group by a decrease in the displacement of the Center of Pressure (CoP) of 6.30 mm and 3.72 mm during single and dual tasking, respectively, compared to an increase in the sham-tDCS group of 0.52 mm and 6.54 mm during single and dual tasking, respectively. A more significant decrease in the walking stride length and average speed were also observed after applying a-tDCS on the M1 compared to sham stimulation. These differences were observed in both simple and dual tasking scenarios.

Similarly, Yosephi et al. [31] observed a significant improvement (reduction) in the posture stability indices, under both static and dynamic settings (with eyes open and closed), in the group that received active stimulation (a-tDCS on M1 combined with posture training) compared to the sham stimulation (posture training + sham tDCS on M1). The average reduction in the static postural stability index in the group receiving a-tDCS applied on M1 was 0.48 degrees and 0.74 degrees with eyes open and closed, respectively, while the sham tDCS group showed a reduction of 0.10 degrees and 0.35 degrees with eyes open and closed, respectively. In terms of dynamic posture stability, the average reduction was 0.69 degrees and 0.60 degrees (eyes open and closed, respectively) in the a-tDCS group, and 0.24 degrees and 0.10 degrees (eyes open and closed, respectively) in the sham tDCS group. The improvement in the BBS score was also significantly higher in the a-tDCS group (9.17 points) compared to both the sham tDCS (0.25 points) and control (0.27 points) groups.

Manor et al. 2016 [29] assessed stability while performing a dual task and observed a significant reduction in the execution cost for both the area and speed of posture sway, as well as an increase in gait speed. The post hoc analysis showed that for each of these outcomes, the cost associated with dual tasking was lower after applying active tDCS versus sham tDCS, and also compared with either baseline condition. Additionally, applying a-tDCS reduced the costs of walking while solving mathematical tasks (SSST). Similarly, the error rate in the SSST during walking trials was found to be lower after applying active tDCS versus sham tDCS, and also compared to both baseline conditions.

Another trial [28] observed a reduction in execution costs during the performance of a dual task that was significantly greater than that in the sham tDCS group. The postural sway speed while standing (mm/s) decreased by 18% (SD = 30) in the a-tDCS groups versus an increase of 25% (SD = 28) in the sham tDCS group. The mean area sway at standing (mm^2^) decreased by 36% (SD = 35) in the a-tDCS group versus 18% (SD = 27) in the sham tDCS group, a change that persisted at the 2 weeks follow-up. Additionally, a significant improvement in the stride time (s) while performing dual tasks was observed in the a-tDCS group compared to sham stimulation, with a reduction of 8% (SD = 9) versus 3% (SD = 10), respectively. However, no significant differences were observed in the TUG test.

Similarly, Zhou et al. 2015 [32] found greater improvements in stability at standing while performing dual tasks in the tDCS group versus the placebo group, with tDCS reducing the impact of performing a cognitive task on the stability of the CoP. Post-hoc analyses revealed that the dual-task cost was lower following active tDCS (1.6%, SD = 31.8) than that after applying sham tDCS (−14.8%, SD = 23.3).

Finally, the study by Zhou et al. 2018 [33] observed a more significant reduction in vibratory thresholds (i.e., greater improvement of vibratory somatosensation) of both foot soles compared to sham stimulation. The average decrease in vibratory somatosensation was 21% (SD = 20) for the right foot sole and 16% (SD = 20) for the left foot sole. The average percent decrease in the time needed to complete the TUG test following tDCS was 6.1% (SD = 6.0; 0.3 ± 0.3 s), while the percentage change following sham stimulation was negligible (0.5 ± 7.1%, or 0.01 ± 0.4 s), although the outcomes did not reach statistical significance (*p* = 0.07). Participants who experienced greater percentage decreases in the vibratory somatosensation of the foot sole also exhibited greater percentage improvements in the TUG time.

#### 3.4.2. Effect on Fine Manual Dexterity

Ljubisavljevic et al., 2019 [27] applied tDCS bilaterally with the anode on the left DLPFC and observed a more significant reduction in dual-task costs versus sham as measured by both SSST (−0.13; 95% CI: −0.18 to −0.06) and GPT (−0.06; 95% CI: −0.10 to −0.009). The effect was more pronounced for the SSST than for the manual dexterity task. Thirty minutes after applying tDCS, the amount of numbers subtracted or pegs inserted did not significantly differ among montages, except for the number of pegs (manual dexterity task) following a-tDCS (left DLPFC). On the other hand, the stimulation montage had no substantial effect on SRT, suggesting that tDCS did not influence SRT, irrespective of the montage of DLPFC stimulation.

#### 3.4.3. Muscle Strength of Elbow Flexors

Oki et al. [30] evaluated muscle strength and did not find significant differences in the average strength (*p* = 0.87) or peak strength (*p* = 0.81) of the biceps brachii (measured on the non-dominant arm) after applying a-tDCS versus sham tDCS. Similarly, the active a-tDCS group did not experience a significant effect on the electromyography amplitude compared to the sham tDCS group (*p* = 0.88).

#### 3.4.4. Effect on Cognitive Function

Manor et al. 2016 [29] observed that the tDCS intervention produced more significant improvements in the MoCA score compared to sham stimulation, and that these results persisted over at least two weeks. The analysis of the MoCA scores showed that those receiving a-tDCS exhibited greater performance increases (8 points, SD = 17) compared to the sham tDCS group (0 points, SD = 8). This improvement only occurred for the visual-space execution function.

## 4. Discussion

The results analyzed in this systematic review show that a-tDCS appears to have positive effects on balance and posture control [25,28,29,31,32] and reduce dual tasks costs [27,28,29,32] in older adults when compared to sham stimulation. Only two trials [26,30] did not report greater improvements in the assessed variables (strength [30] and balance [26]) in the intervention group compared with sham stimulation. However, the great variability observed in the measurement tools hinders the withdrawal of firm conclusions. Anodal stimulation of the primary motor area (M1) [25,26,30,31] and anodal stimulation of the DLPFC [27,28,29,32] were the most common protocols for all the included studies. Trials applying stimulation on the DLPFC [27,28,29,32] evaluated the execution cost of performing dual tasks and all found that the effect of tDCS was superior to that of sham stimulation. Most studies [25,28,29,30,32,33] conducted protocols of intervention with tDCS applied offline. This was not the case of Ljubisavljevic et al., 2019 [27] and Yosephi et al., 2018 [31], where participants performed dual tasks while applying the intervention (online stimulation) and observed a greater performance during the bilateral stimulation versus unilateral stimulation. This outcome seems to support online stimulation with tDCS as an effective treatment for reducing dual task costs, although additional studies are needed to compare the effects of online versus offline interventions. On the other hand, Kaminski et al., 2017 [26] applied a single online session of bilateral a-tDCS on the M1 and did not observe significant differences compared to the group receiving postural training. The lack of observed effect in this study could be attributed to the use of low current intensity (0.04 and 0.02 mA/cm^2^), the lowest of any study included in this review, and the delivery of only a single session.

All trials [25,29,31,32,33] obtained significant improvements relative to placebo for any of the variables related to stability and balance. The most pronounced effects were found in the reduction of the impact of dual tasking (posture control and cognitive task) on stability. For example, improvement in balance was observed during the simultaneous performance of two tasks following an intervention with tDCS. Although tDCS appears to positively affect balance, the optimal protocol for improving posture control and stability could not be determined, since different types of interventions reported similar outcomes. More studies are warranted that compare the different protocols of stimulation in order to discern which one yields greater gains in balance. 

With regard to the protocol stimulation parameters, only Ljubisavljevic et al., 2019 [27] compared the effect of tDCS in different locations and montages. Bilateral anodal stimulation of the DLPFC showed greater improvements in manual dexterity than unilateral anodal and unilateral cathodal stimulation applied alone. Future studies should address these stimulation parameters, so that anodal versus cathodal stimulation can be compared at each stimulation site for the study to declare a strong conclusion regarding the stimulation protocol used.

On the other hand, Zhou et al., 2018 [33] conducted a trial aimed at improving the vi-bratory somatosensation of the foot sole and its relationship with stability. Their out-comes were significantly better (lower vibratory somatosensation) in the active tDCS stimulation versus the sham group. This provides preliminary evidence to indicate that non-invasive modulation of the somatosensory cortex can be an effective strategy for improving somatosensation of the foot sole and therefore functionality (gait, balance) in older adults. Even if tDCS was applied on the left cerebral hemisphere, vibratory sensation improved in both feet. The electrode size (35cm^2^) and the subsequent distribution of current can explain this finding, as the unilateral tDCS intervention could have stimulated brain areas related to the processing of sensory information of both hemibodies. In terms of duration of the effect, only Manor et al., 2018 [28] included a follow-up at two weeks after the intervention, showing that the magnitude of the effect persisted at least over the analyzed two weeks. Therefore, future research is needed to assess the effect of tDCS in the intermediate and long term in order to determine the duration of the effect. Of those studies that did not report differences between the achieved improvements in the intervention or placebo groups, Oki et al., 2019 [30] measured the muscle strength of the biceps brachii in the non-dominant arm after a single online stimulation with tDCS. However, the outcomes of this trial are in agreement with current literature, where no consensus on the effect of tDCS on muscle strength has been reached [35,36,37,38,39,40,41]. So far, only one trial has examined the values of muscle strength in young adults (40 years) without relevant pathologies [42], and it did not find any effect on this variable. Therefore, more studies are required in the younger population to determine the impact of tDCS on muscle strength. Similarly, Kaminski et al., 2017 [26] compared active with sham stimulation in adults and did not find significant differences between treatments, which was in contrast with the findings of an earlier study in the younger population [43]. Therefore, the effects of tDCS appear to differ between younger and older adults. These findings could stem from differences in neuronal plasticity between both population segments, with older people showing retarded plasticity in the M1 region [44]. Neuromodulation stimulation studies are also required to compare the effect of the intervention in younger with older adults, especially as only a small number of trials have been published comparing people from both age ranges.

The main limitation of this systematic review was the small sample size and low number of neuromodulation interventions; a previous study showed greater positive effects with a multi-session program of tDCS in older adults [45]. Another limitation of this systematic review was the great variability of the stimulation parameters, experimental design and outcome measures used in a small number of published trials. This high level of heterogeneity of study parameters precluded performing a meta-analysis of the findings. Additionally, more studies are required that evaluate muscle strength as an outcome measures; only one functional study met the inclusion criteria for inclusion in this review [30]. Finally, a better understanding of the neural mechanisms that contribute to the functional effect of tDCS is required. Simultaneous tDCS-fMRI could be an option for identifying the effects of the intervention on specific neural structures in older adult participants [46].

## 5. Conclusions

In view of the analyzed outcomes, stimulation with a-tDCS on the M1 appears to be effective for improving balance and posture control in healthy older adults. Additionally, tDCS applied on the DLPFC can reduce dual task costs and improve the scores of tests evaluating these tasks, as well as improving some gait parameters while performing two tasks simultaneously (cognitive and motor). However, more trials are required to homogenize parameters and determine which protocol yields greater improvements, thus enabling the standardization of protocols of tDCS.

## Figures and Tables

**Figure 1 jcm-10-02981-f001:**
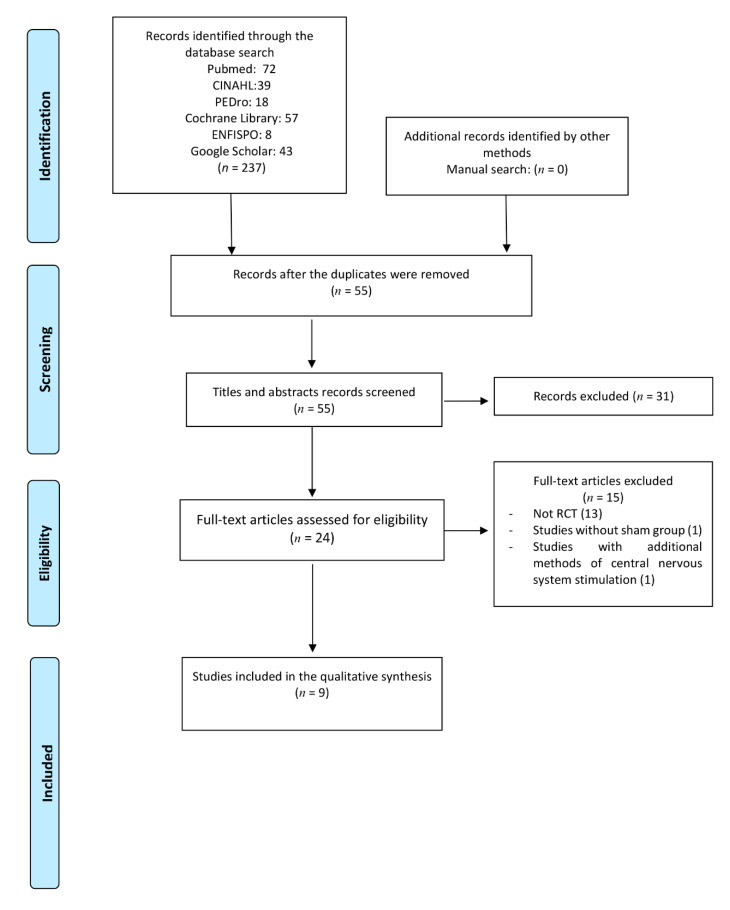
Flow diagram of the systematic review process, including the identification, screening and eligibility steps, recommended by the guidelines included in the Preferred Reporting Items for Systematic Reviews and Meta-Analyses (PRISMA).

**Table 1 jcm-10-02981-t001:** Methodological quality of included articles in accordance with the PEDro scale.

	Baharlouei et al., 2020 [25]	Kaminski et al., 2017 [26]	Ljubisavljevic et al., 2019 [27]	Manor et al., 2016 [28]	Manor et al., 2018 [29]	Oki et al., 2019 [30]	Yosephi et al., 2018 [31]	Zhou et al., 2015 [32]	Zhou et al., 2018 [33]
Eligibility criteria	0	0	1	0	1	1	0	0	0
Randomized allocation	11	1	1	1	1	1	1	1	1
Concealed allocation	1	0	0	0	1	0	0	0	0
Basal intergroup similarities	1	1	1	1	1	0	1	1	1
Blinding of participants	1	1	1	1	1	1	1	1	1
Blinding of therapists	0	0	0	1	1	1	0	1	1
Blinding of assessors	1	0	1	0	0	0	1	0	0
Follow-up	1	0	1	1	1	1	1	1	1
Intention-to-treat analysis	1	1	1	1	1	1	1	1	1
Between-group statistical comparison	1	1	1	1	1	1	1	1	1
Point measures and variability measures	1	1	1	1	1	1	1	1	1
Total Score	9/10	6/10	8/10	8/10	9/10	7/10	8/10	8/10	8/10

**Table 2 jcm-10-02981-t002:** Characteristics of transcranial direct current stimulation (tDCS).

Study	Main Electrode	Electrode Size (cm^2^)	Intensity (mA)	Current Density (mA/cm^2^)	Location of Stimulation	Duration (min) and Mode	Number of Sessions (Washout Period)
Baharlouei et al., 2020 [25]	Anode	a: 27 c: 36	2	a: 0.07 c: 0.05	M1	20 (offline)	2 (1 week)
Kaminski et al., 2017 [26]	Anode	a: 25 c: 50	1	a: 0.04 c: 0.02	M1 Bilateral	20 (online)	1
Ljubisavljevic et al., 2019 [27]	Anode Bilateral	35	1.5	0.04	DLPFC	30 (10 min online)	2 (2 weeks)
Manor et al., 2016 [28]	Anode	35	2	0.06	DLPFC	20 (offline)	2 (1 week)
Manor et al., 2018 [29]	Anode	35	2	0.06	DLPFC	20 (offline)	10 (5 s/week)
Oki et al., 2019 [30]	Anode	35	1.5	0.04	M1	20; 17ʹ30ʺ(offline)	3 (10 days)
Yosephi et al., 2018 [31]	Anode	35	2	0.06	M1	20 (online)	6 (3 s/week)
Zhou et al., 2015 [32]	Anode	35	2	0.06	DLPFC	20 (offline)	2 (1 week)
Zhou et al., 2018 [33]	Anode	35	2	0.06	Sensorimotor cortex	20 (offline)	2 (1 week)

a: anode. c: cathode. DLPFC: dorsolateral prefrontal cortex. M1: motor cortex.

**Table 3 jcm-10-02981-t003:** Main results.

Study	Number of Participants	Age	Study Design	Outcome Measures	Measurement Time Point	Results (Versus Sham tDCS)
Baharlouei et al., 2020 [25]	32	67.6 (6.3)	Double-blinded, sham-controlled, crossover study	CoP displacement, stride length, walking speed	Before and after	↓ CoP, stride length and velocity
Kaminski et al., 2017 [26]	30	67.7 (6.0)	Single-blinded, sham-controlled, randomized crossover study	Time in balance	Before and after	NS
Ljubisavljevic et al., 2019 [27]	22	62.6 (3.2)	Double-blinded, sham-controlled, crossover study	Manual dexterity (GPT), Cognitive task (SSST), Dual-task, simple reaction time	Before, during, and after	↓ Dual-task costs during bilateral tDCS
Manor et al., 2016 [28]	37	61 (5.0)	Single-blinded, sham-controlled, randomized crossover study	Dual-task costs while standing, walking, and serial subtraction performance	Before and after	↓ Dual-task costs while standing and walking
Manor et al., 2018 [29]	19	80.5 (4.0)	Double-blinded, sham-controlled, randomized parallel study	MoCA, TUG, dual-task cost, walking speed, sway speed	Before, after, and 2 weeks later	↑ MoCA, ↓ Dual task standing postural sway, ↓ Stride time dual-task
Oki et al., 2019 [30]	11	85.8 (4.3)	Double-blinded, sham-controlled, randomized crossover study	Isometric maximal contractions, EMG activity	Before and after	NS
Yosephi et al., 2018 [31]	65	66.1 (4.0)	Double-blinded, sham-controlled, randomized parallel study	Berg Balance Score (BBS), stability indices	Before and after	↑ Postural stability indices and BBS scores
Zhou et al., 2015 [32]	20	63 (3.6)	Double-blinded, sham-controlled, randomized crossover study	CoP fluctuations	Before and after	↑ Complexity of standing postural sway
Zhou et al., 2018 [33]	20	61 (4.0)	Double-blinded, sham-controlled, randomized crossover study	Standing vibratory threshold (SVT), TUG	Before and after	↓SVT in both soles

CoP: center of pressure, EMG: electromyography, MoCA: Montreal Cognitive Assessment, NS: non-significant, TUG: Timed Up-and-Go Test, SSST: Serial Sevens Subtraction Test; GPT: Grooved Pegboard Test; ↑: increase; ↓: decrease.

## Data Availability

Not applicable.

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
