# Peer review of "Can Transcranial Direct Current Stimulation Enhance Functionality in Older Adults? A Systematic Review"

_jcm, 2021, doi:10.3390/jcm10132981_

Round 1
Reviewer 1 Report
BRIEF SUMMARY
This systematic review aimed at compiling and summarizing the currently available scientific evidence about the effect of tDCS on functionality in older adults over 60 years of age. Authors conclude that stimulation with a-tDCS appears to be effective for improving balance and posture control in healthy older adults, and can reduce dual task costs and better the scores of tests evaluating these tasks, as well as improve some gait parameters while performing two tasks simultaneously.
I congratulate authors on their work. The topic is timely and clinically important. My main concern is the quality of reporting. My initial suggestions are listed below. The article has a potential but can only be judged once the initial concerns are taken care of by the authors.
MAJOR COMMENTS
- Please structure your methods according to PRISMA reporting guidelines. Methods are very superficial for this to be regarded high quality systematic review. In the current form, the paper is not ready to be peer-reviewed.
- Why meta-analysis was not performed? No justification is provided. 9 RCTS is enough to perform meta-analysis and increase confidence in the findings.
Author Response
BRIEF SUMMARY
This systematic review aimed at compiling and summarizing the currently available scientific evidence about the effect of tDCS on functionality in older adults over 60 years of age. Authors conclude that stimulation with a-tDCS appears to be effective for improving balance and posture control in healthy older adults, and can reduce dual task costs and better the scores of tests evaluating these tasks, as well as improve some gait parameters while performing two tasks simultaneously.
I congratulate authors on their work. The topic is timely and clinically important. My main concern is the quality of reporting. My initial suggestions are listed below. The article has a potential but can only be judged once the initial concerns are taken care of by the authors.
We thank this Reviewer for their positive comments and appreciation of our study.
MAJOR COMMENTS
1.Please structure your methods according to PRISMA reporting guidelines. Methods are very superficial for this to be regarded high quality systematic review. In the current form, the paper is not ready to be peer-reviewed
We agree with the reviewer on this point. We have rewritten `Methods Section’ according to your suggestion.
In the new version of the manuscript, we have added subsections in methods, that are under the PRISMA guidelines. This change in the structure will increase the quality of our work.
2.Why meta-analysis was not performed? No justification is provided. 9 RCTS is enough to perform meta-analysis and increase confidence in the findings.
Thank you for your appreciation. Although it is true that nine RCTs is enough to perform a meta-analysis, due to the high degree of heterogeneity among studies and the lack of consensus in the outcomes reported, this option was ruled out. A descriptive qualitative analysis of the data was performed instead. Hopefully, in the near future it would be possible to perform a meta-analysis in order to increase the evidence about tDCS.
In the new manuscript version we have added this point as a limitation.

Reviewer 2 Report
The review was well-written, thorough and covering an important topic in brain stimulation. Specifically, the use of tDCS in older adults to manage cognitive and physical decline. I have a few recommendations for improving the manuscript that the authors may wish to consider.
Some further discussion of the study designs of the reviewed studies is warranted. For example, how many included an active control site or using contrasting stimulation (e.g. cathodal stimulation) at the same stim site? These studies will allow dissociable, causal relationships to emerge that provide greater evidence than anodal v sham at a single stimulation site. This design will also allow for a strong conclusion like the review finishes with, that M1 effective in improving balance and dlPFC for reducing dual-task costs.
The majority of the studies reviewed are single or 2-3 sessions. As tDCS may exert different and more long-lasting effects when used in a multisession fashion (see Perceval et al, 2020, Brain & Language), this should be noted as a potential future direction for this area of brain stimulation research.
Simultaneous tDCS-fMRI is also a possibility in older adults (see Martin et al, 2017, Journal of Cognitive Neuroscience) and could be discussed as another future direction to understand how tDCS directly affects neural networks related to dual-task costs or balance. Likewise, using fMRI to predict effects of tDCS may also be a valid future direction the review would like to mention.
Some further cautionary notes on how variability in brain ageing, including differences in cognitive decline would benefit the review. Moreover, a cautionary note about translating results found in young adults to older adults is warranted as research is scarce on directly comparing the two age groups.
I think overall, a paragraph on future directions for this specific field would benefit the discussion.
Author Response
REVIEWER 2
The review was well-written, thorough and covering an important topic in brain stimulation. Specifically, the use of tDCS in older adults to manage cognitive and physical decline. I have a few recommendations for improving the manuscript that the authors may wish to consider.
We would like to thank the reviewer for the constructive review, the positive comments and appreciation of our work. We believe that the proposed changes will make our manuscript more robust. The reviewer’ comments (in bold) and detailed responses to each point of concern are listed below. Changes in the new manuscript are highlighted in yellow.
Some further discussion of the study designs of the reviewed studies is warranted. For example, how many included an active control site or using contrasting stimulation (e.g. cathodal stimulation) at the same stim site? These studies will allow dissociable, causal relationships to emerge that provide greater evidence than anodal v sham at a single stimulation site. This design will also allow for a strong conclusion like the review finishes with, that M1 effective in improving balance and dlPFC for reducing dual-task costs.
The majority of the studies reviewed are single or 2-3 sessions. As tDCS may exert different and more long-lasting effects when used in a multisession fashion (see Perceval et al, 2020, Brain & Language), this should be noted as a potential future direction for this area of brain stimulation research.
Simultaneous tDCS-fMRI is also a possibility in older adults (see Martin et al, 2017, Journal of Cognitive Neuroscience) and could be discussed as another future direction to understand how tDCS directly affects neural networks related to dual-task costs or balance. Likewise, using fMRI to predict effects of tDCS may also be a valid future direction the review would like to mention.
Some further cautionary notes on how variability in brain ageing, including differences in cognitive decline would benefit the review. Moreover, a cautionary note about translating results found in young adults to older adults is warranted as research is scarce on directly comparing the two age groups.
I think overall, a paragraph on future directions for this specific field would benefit the discussion.
We agree with the reviewer on this point. We have now included two paragraphs in the discussion section as suggested addressing each point. We have now highlighted the potentials future directions of tDCS in older adults.